# Tuning the Wavelength: Manipulation of Light Signaling to Control Plant Defense

**DOI:** 10.3390/ijms24043803

**Published:** 2023-02-14

**Authors:** Susan Breen, Hazel McLellan, Paul R. J. Birch, Eleanor M. Gilroy

**Affiliations:** 1Division of Plant Sciences, University of Dundee, At James Hutton Institute, Errol Road, Invergowrie, Dundee DD2 5DA, UK; 2Cell and Molecular Sciences, James Hutton Institute, Errol Road, Invergowrie, Dundee DD2 5DA, UK

**Keywords:** plant, light, immunity, pathogen effectors, signaling, pathogenicity

## Abstract

The growth–defense trade-off in plants is a phenomenon whereby plants must balance the allocation of their resources between developmental growth and defense against attack by pests and pathogens. Consequently, there are a series of points where growth signaling can negatively regulate defenses and where defense signaling can inhibit growth. Light perception by various photoreceptors has a major role in the control of growth and thus many points where it can influence defense. Plant pathogens secrete effector proteins to manipulate defense signaling in their hosts. Evidence is emerging that some of these effectors target light signaling pathways. Several effectors from different kingdoms of life have converged on key chloroplast processes to take advantage of regulatory crosstalk. Moreover, plant pathogens also perceive and react to light in complex ways to regulate their own growth, development, and virulence. Recent work has shown that varying light wavelengths may provide a novel way of controlling or preventing disease outbreaks in plants.

## 1. Introduction

Plants are sessile organisms that need to optimize their responses to the environment in which they occur. How plants decide where and when to allocate limited resources can determine whether the whole plant survives. Stress-inducing factors include light quality and intensity, temperature, salt concentration, water and nutrient availability, as well as interactions with other organisms such as a wide array of microbiota, pathogens, and pests [1]. Each unique organism perceives and differentially responds to light wavelengths [2,3,4]. This is mainly due to differences in each organism’s exclusive chemical composition and the environmental niche in which they have evolved and finely tuned their developmental processes to suit their lifestyle. In addition, most organisms perceive light using numerous mechanisms to help them sense and respond appropriately to the presence, absence, photoperiod, quality, intensity, and timing of available light received in any given environment [3,5]. FAO reports that climate change is adversely affecting most biological systems at multiple scales, from genes to ecosystems [6]. Furthermore, modern agricultural practices, such as the global cultivation of plants in diverse geographical regions and in high-density monocultures, has generated conditions that can weaken plant defenses, decreasing environmental resilience and increasing disease and pest severity. Exacerbating this situation is the financial pressures associated with the increasing occurrence of extreme climatic events, pushing grower and political demand for low-input/high-resilience crops that perform predictably under diverse stresses. How crops may respond could be better anticipated with a fuller understanding of how the environment impacts plant immunity and growth and development.

Advances over the past two decades have developed an overview of defense regulation in plants and the interactions of a number of essential hormonal players [7,8]. This provides a framework by which to interpret the molecular relationships induced under environmental variables (such as light) and how these may influence the major regulatory hubs that orchestrate plant immunity [9]. In this review, we will briefly summarize the key players in plant immunity and concentrate on the ways in which light may influence them through crosstalking regulatory hubs and the various ways in which pathogens respond to light and manipulate host responses to cause disease. We conclude by highlighting some of the major questions to be answered in the field of plant–microbe research and suggest some of the key areas that are in greatest need of further research investment.

## 2. Immunity and Light Signals

### 2.1. Plant Immunity

Plants interact with a wide range of organisms and are wired genetically to perceive threats, welcome allies, and fine-tune inducible responses. This shapes ecological processes, plant health, and crop resilience in any given environmental niche. In a recent FAO report, it is estimated that around 40% of the global crop production is lost to pests each year and that plant diseases cost the global economy over USD 220 billion [6].

Plants contain multiple systems to recognize the presence of potential invaders. Plants perceive cell damage (damage-associated molecular patterns, DAMPs) and detect exposed and highly conserved molecules from different classes of microbe/pathogen-associated molecular patterns (MAMPs/PAMPs). These damage and non-self molecules, as well as some apoplastic effectors, are recognized by plasma-membrane-localized pattern-recognition receptors (PRRs), inducing a heightened state of immunity known as PAMP-triggered immunity (PTI) which is well reviewed [8,10,11,12]. Pests and microbes that have coevolved with their hosts are able to secrete effectors into host tissues rapidly enough to suppress PTI and allow infection. This is a compromised state known as effector-triggered susceptibility (ETS) [8,13]. However, pathogen effectors are also subject to detection by the host via membrane PRRs and intracellular receptors belonging to the nucleotide-binding site leucine-rich repeat (NBS-LRR) family, known as NLRs or R (resistance) proteins. An effector can be either recognized directly, through protein interactions, or indirectly, by exploitative activities on particular guarded host targets in a process known as effector-triggered immunity (ETI), which is also very well reviewed [8,12,13]. 

Briefly, both PTI and ETI involve similar signaling molecules that transmit cell-to-cell communications, such as reactive oxygen species (ROS) accumulation, Ca2+ influx, and nitric oxide (NO) production [12,14]. The induced signaling cascades, during both PTI and ETI, are regulated by a range of protein oxidases, kinases, phosphatases, and ubiquitin ligases to name but a few. These signaling cascades promote local physiological responses to impede invaders such as callose deposition, stomatal closure, and ethylene and SA biosynthesis. Ultimately, signaling leads to large-scale transcriptional reprogramming via activation of defense-related transcription factors (such as WRKY, bZIPs, and BHLHs) that promote the generation of antimicrobial components to fight pathogen attack [15]. The biosynthesis of the immune signal salicylic acid (SA) occurs in various subcellular compartments, including the cytosol and chloroplasts [16]. Moreover, the chloroplast’s role in systemic acquired resistance (SAR) now includes being the site of biosynthesis of lysine-derived pipecolic acid (Pip) [17]. Furthermore, the downstream-localized FLAVIN-DEPENDENT MONOOXYGENASE 1 (FMO1) that converts Pip to the mobile SAR signal, N-hydroxypipecolic acid (NHP), is also thought to localize to the chloroplast [18]. NHP works in collaboration with SA to promote a long-lasting layer of broad-spectrum protection in distal tissues [17,18,19]. Both ETI and some PTI responses evoke a localized form of programmed cell death aimed to confine biotrophic pathogens that require living tissue at the site of infection, with ETI being considered to proceed with more speed and intensity. PTI has recently been shown to be involved in potentiating ETI responses, which can, in turn, reinforce PTI [8,12,14,20].

The phytohormone jasmonic acid (JA) plays a central role in plant defense responses against necrotrophic fungal pathogens and insect feeding and is usually considered to be antagonistic to SA-driven responses [21]. The JA signaling pathway’s functions in defense, growth, and development is well characterized and reviewed [22,23]. Briefly, JA perception leads to the activation of signaling cascades that activate key transcription factors (e.g., NAM, ATAF1/2, and CUC2 (NACs)) and the master coordinator basic helix–loop–helix (bHLH), MYELOCYTOMATOSIS (MYC2) [24]. MYC2 is essential for induced systemic resistance (ISR) triggered by beneficial soil microbes [25]. MYC2 is central for regulating crosstalk between JA and abscisic acid (ABA), SA, gibberellins (GAs), and auxin (IAA) signaling pathways [26]. JA biosynthesis begins at the inner chloroplast membrane and is known to be affected by light [27,28]. Furthermore, there are many environmental and developmental cues that induce various spatially and temporally regulated phytohormone pathways which are known to intersect in different ways with plant immunity [7].

Some subcellular organelles have been shown to respond to environmental conditions which can affect the host plant’s ability to induce the appropriate defenses. Chloroplasts were historically considered to be responsible for the production of sugars from water, CO_2_, and sunlight during photosynthesis, and mitochondria produce energy from the breakdown of sugar during respiration. The role of chloroplasts in plant immunity has been of recent interest [7,29,30,31]. Chloroplasts are mobilized during pathogen infection to cellular regions sensitive to invader attack and to the nucleus, presumably to optimize the rapid exchange of signals, nucleic acids, and proteins [29,32]. Chloroplasts have also been demonstrated to be important for phytohormone production and the synthesis of secondary metabolites, ROS, and nitric oxide that all have antimicrobial and signaling properties involved in PTI and ETI immunity [31]. There appears to be mounting evidence that light signaling and light-responsive organelles, particularly chloroplasts, play a dual role in the allocation of resources for growth and plant immunity.

### 2.2. Light Sensing in Plants 

Light is essential for the life of plants. The wavelengths of light with maximum photosynthetic activity are violet blue and red light (BL and RL), and the energy in the light reaching a plant can vary in predictable ways in terms of the time of day, season, and latitude as well as transiently by changes in cloud cover, the topography of the landscape, and shade from neighboring plants [1,4,33]. Therefore, plants in their native environments have adapted to respond appropriately to the range of distinguishable light signals by modifying photosynthetic rates and developmental processes, including germination, de-etiolation, stomatal development, circadian rhythm, and flowering time to suit their environment and adaptations [3,9]. In response to more rapidly changing environmental factors, plants can adapt their physiology, for example, via chloroplast movement, leaf positioning, and stomatal opening [4].

Plants possess a range of photoreceptors to detect different wavelengths of light, particularly for BL and RL. Plants can sense the intensity, duration, quality, and direction of light with chlorophyll-containing tissues [1]. Plants detect red/far red light (RL/FRL) using phytochromes [34] and ultraviolet B (UV-B) using UV resistance locus 8 [35]. However, there are several types of receptors associated with BL perception including cryptochromes [36,37], phototropins (phot1 and phot2 [38,39,40]), and the Kelch-containing F-box protein (KFB) subfamily [38,41] (Figure 1).

There are various mechanisms by which photoreceptors relay light cues to the immune pathways, but many are not well understood. Evidence is increasing that light spectral changes can modulate the induction of plant immune responses against pests and pathogens [1,42,43]. Several hypotheses predict that this is for better allocation of resources at times when growth needs to be prioritized to avoid shading or to preserve tissues that are functioning at optimum light energy capturing capacity. How inputs from each photoreceptor fuse to promote signaling for optimal growth responses and are balanced with appropriate resource allocation for plant immunity will be key to developing cultivars that are better adapted to their environment.

### 2.3. Red Light Regulates Immunity

Phytochromes are the principal photoreceptors involved in red (RL) and far-red light (FRL) perception (R:FR ratio). PhyA mediates various plant responses to FRL, but phytochrome B (phyB) plays the most prominent and well-characterized role. Under high planting density and leaf shading conditions, RL is preferentially absorbed by chlorophyll, reducing the R:FR ratio and the proportion of phyB in the active (Pfr) form [44]. The depletion of Pfr form is used for reliable early detection of competing neighbors and activates the escape strategy, known as the shade avoidance syndrome [45] (Figure 1).

PhyB in its active Pfr form is primarily localized in the nucleus under high R:FR ratios. Active phyB suppresses the accumulation and activity of shade-avoidance-associated, growth-promoting bHLH transcription factors (TFs) known as Phytochrome-Interacting Factors (PIFs) [46,47]. Nuclear accumulation of light-activated phyA requires FAR-RED ELONGATED HYPOCOTYL1 (FHY1) and FHY1-LIKE (FHL) [48] (Liu and Wang, 2020). In addition, the first known positive regulator of light, the basic leucine zipper (bZIP) transcription factor, HY5, plays a role in transcriptional regulation downstream of RL and FRL, as well as in BL and UV-B perception [49]. HY5 positively regulates the phyA-mediated inhibition of hypocotyl elongation and is thought to interact with phyB to promote PCD in response to RL. Changes in R:FR also trigger differential responses in the signaling pathways of phytohormones, such as auxin, gibberellins (GAs), ethylene, and brassinosteroids (BRs) [47]. Growth-promoting BRs, known to antagonize JA-mediated responses and modulate the strength of PTI-based responses, increase in the shade [48]. Under low R:FR ratios, phyB is converted to its inactive from (Pr), increasing PIF4, PIF5, and PIF7 activity, leading to increased expression of auxin biosynthesis genes and induction of morphological changes for shade avoidance, including elongation responses in Arabidopsis [50,51]. Furthermore, GAs increase in low R:FR ratios, promoting growth by increasing the turnover of DELLA proteins (known repressors of PIFs and activators of JA signaling [52]). In addition, bZIP transcription factors activated in light by CRYs, phyB, and hormone pathways (e.g., BR-activated BRASSINAZOLE-RESISTANT 1 (BZR)) are known to heterodimerize, providing regulatory crosstalk at the transcription level [53,54]. Intriguingly, BRs play a significant role in light-induced regulation of plant development, but conversely, light does not appear to have a meaningful effect on BR levels or the accumulation of BZR1 [55]. 

The effect of RL and FRL on plant immunity has been well studied, but many of the underlying molecular mechanisms remain to be elucidated [56]. In response to phyB inactivation, plants are mainly more susceptible to pathogens and herbivores [42,57]. For example, tomato phyB positively regulates defenses against the herbivorous insect *Spodoptera eridania*, and both Arabidopsis phyA and phyB contribute to defense responses against the incompatible bacterial strain *Pseudomonas syringae* pv. *tomato* (*Pto*) DC3000 [56]. The main impact of RL on plant defense has been linked to JA sensitivity and JASMONATE ZIM (JAZ) activity that mediate responses to necrotrophic pathogens and herbivory. JA has been demonstrated to regulate phyA-mediated FR signaling. Recently, it has been found that JAZ1 represses FHY3, a transposase-derived transcription factor, which regulates FHY1/FHL gene expression and consequently decreases phyA in the nucleus [48]. Furthermore, phyB is required for the light-induced expression of JA biosynthesis genes [56]. In addition, phyB-dependent light signaling enhanced plant defense responses by positively regulating the concentration of JA and phyB functions coordinately with the JA pathway to control plant defense response against the necrotrophic fungus, *Botrytis cinerea* [56]. The inactivation of phyB during shade avoidance syndrome causes increased stability of JAZ repressors, enhanced degradation of the bHLH MYC TFs, and reduced plant sensitivity to JA [45,52,58,59]. MYC2 regulates interactions between JA signaling and light, phytochrome signaling, and the circadian clock [60]. MYC2 mediates JA-regulated plant development, lateral and adventitious root formation, flowering time, and shade avoidance syndrome which vary depending on the plant species [26]. Recently, a sulfotransferase (ST2a) that plays a role in reducing the pool of active jasmonates was found to be strongly upregulated by shading via PIF signaling [5]. The reduction in JA signaling in Arabidopsis could promote DELLA degradation and JAZ stability [52] and differential regulation of MYC TFs and their JAZ repressors [59]. Overall, there is significant evidence that shade-intolerant plants activate phytochrome-mediated light signaling and that this plays an important role in regulating JA-mediated defenses [5].

Interestingly, SA-mediated signaling pathways are upregulated in high R:FR ratios, increasing resistance to biotrophic pathogens [61,62]. Exposing plants to RL has been shown to increase levels of SA and induce SA signaling, which in turn promotes the production of ROS [62]. The light-regulated TF, HY5, positively regulates ROS production and cell death and binds to G-box (CACGTG) elements found in many ROS-, hormone- and defense-responsive gene promoters [26,49]. However, whether HY5 is involved in promoting cell death through ROS and SA signaling in red light remains to be fully elucidated. Moreover, typically antagonistic signaling induced by phyA and phyB has been shown to contribute to defense responses against *Pto* DC3000 through interactions with the pathogen/SA-mediated signal transduction pathway [63]. Conversely, low R:FR conditions, which reduce the levels of active phyB, repress SA-induced defenses [50]. This causes major downregulation of SA-responsive genes controlled by the reduced phosphorylation and relocalization of non-expressor of pathogenesis-related genes 1 (NPR1) [61]. Photoreceptor *phyB* mutants are significantly more susceptible to *Pto* DC3000 than Col-0 WT. In addition, the silencing of *NPR1* partially compromises red-light-induced resistance against *Pto* DC3000, suggesting it plays an important role in this response [61]. RL-regulated mechanisms appear to involve both JA and SA, e.g., in tobacco defense against cucumber mottle virus [64] and in rice developmentally controlled resistance to the blast fungus *Magnaporthe grisea* [65]. On the other hand, phyB has also been shown to inhibit BZR1 signaling to negatively regulate resistance to sheath blight in rice [66]. In one further example, RL could significantly suppress gall formation and root knot nematode *Meloidogyne incognita* (RKN) development and caused transient upregulation of *PR1* and *WRKY70* transcripts in infected plants [67]. The RL-induced systemic defense was attributed to both increased JA and SA and the transcript levels of their biosynthetic genes in roots [67]. 

Overall, it can be concluded that there is significant evidence that RL and FRL wavelength perception regulates biotic stress responses in plants through the control of photomorphogenesis and shade avoidance responses, hormone dynamics, and the regulation of transcription factors that relay and fine-tune adaptations to environmental conditions [4,42]. 

### 2.4. Blue Light Regulates Immunity

There are far fewer publications regarding the role of BL perception in regulating plant immunity. We discuss the effects of BL on plant immunity by examining each different family of blue light receptor.

#### 2.4.1. Phototropins

Phototropins (phots) trigger numerous light-capturing physiological responses to optimize photosynthetic productivity by regulating stomatal opening [68], chloroplast movements [69], and phototropism [39,70]. Phots are members of the AGC (cAMP-dependent protein kinase A, cGMP-dependent protein kinase G, and phospholipid-dependent protein kinase C) family of kinases. Phots contain two N- terminal light–oxygen–voltage (LOV) domains (LOV1 and LOV2) that, when activated by BL, regulate the C-terminal Ser/Thr kinase domain which is essential for conformational changes, autophosphorylation of the phototropins, and phosphorylation of downstream substrate proteins [38,71]. So far, there have been only a handful of phosphorylated substrates of phots, with the most significant being interactions with the BTB/POZ-domain-containing NRL family members [72]. NRLs promote associations with the Cullin-RING E3 ubiquitin ligase complex (CUL3) to target substrate proteins for proteasomal degradation, including the positive regulator of plant immunity, SWAP70 [73]. In addition, more than one NRL family member has been linked to chloroplast movement which may also influence the migration and signaling of chloroplasts during defense responses [74]. In support of this, BL and the transient overexpression of potato phots was found to significantly increase the susceptibility of *Nicotiana benthamiana* to the late blight pathogen, *Phytophthora infestans* [75]. Furthermore, phot signaling increases cytosolic Ca^2+^ concentrations in BL responses, which will have a significant influence on calcium-regulated processes, including organelle movements and transcription factor activation [76]. Intriguingly, there is far less evidence supporting phot-regulated gene expression [72], as is common downstream of the activation of many other photoreceptors (Figure 1). 

Could phot-regulated chloroplast movement significantly moderate the speed and timing of immune signaling and deployment of appropriate defenses? Chloroplasts play crucial roles as mobile organelles that produce stress-related phytohormones, ROS, and other stress signals under detrimental environmental conditions [30,77,78]. The ROS that chloroplasts overproduce are associated with the spatiotemporal regulation of death or defense signaling. Furthermore, the chloroplast and the nucleus appear to have critical interconnected retrograde signaling pathways initiated by ROS and ROS-modified target molecules to preserve chloroplast integrity and whole-cell-level functions [78]. In addition, chloroplasts migrate towards the site of attempted infection to amplify the speed of subcellular communications at times when major transcriptional changes are induced [29]. So, although phot signaling modulates multiple physiological and developmental processes that are typically associated with optimizing the plant’s ability to capture light energy for photosynthesis, the induced changes in stomatal opening, Ca^2+^ signaling, phosphorylation, ubiquitination, and organelle movements will all have a significant impact on the speed and type of plant defenses induced under varying BL conditions. 

#### 2.4.2. Cryptochromes (CRYs)

CRYs are flavoprotein photoreceptors that occur in all plant species and are characterized by an N-terminal domain, a photolyase homology region (PHR), and a disordered C-terminal tail [79]. In Arabidopsis, low-intensity BL perceived by cryptochromes (CRY1 and CRY2) is a trigger for shade avoidance responses, promoting elongation and leaf upward positioning, acting via PIF4 and PIF5 (Figure 1). In addition, CRY-mediated floral initiation has been well studied in Arabidopsis. CRYs are inactive monomers in the dark and homodimerize in response to BL. Photoactivated CRYs modulate transcription via two distinct processes: indirectly by inactivating the COP1/SPA E3 ligase complex which leads to the promotion of TF activity, or directly by increasing affinity to bHLH TF families by physical interaction with CIBs (CRY-INTERACTING bHLH), PIFs (AUX/IAA (Auxin/INDOLE-3-ACETIC ACID)), and the COP1-SPA (CONSTITUTIVE PHOTOMORPHOGENESIS 1-SUPPRESSORS OF PHYTOCHROME A) complexes [37]. In addition, CRYs can bind other photoreceptors such as phytochromes and Zeitlupe (ZTL) F-box proteins [80]. These light-dependent CRY interactions significantly modulate gene expression in response to multiple signals and therefore crosstalk with multiple inducible systems, including plant immunity. Furthermore, photoexcitation of CRYs modifies their affinity to regulatory proteins, such as BICs (blue-light inhibitors of CRYs) and PPKs (photoregulatory protein kinases), forming additional regulation of cry activity or abundance in various light environments.

CRY1 has been shown to promote R-protein-mediated resistance and *PR* gene expression in Arabidopsis in response to challenge with *Pto* DC3000 carrying AVRRpt2 [81]. In the same year, CRY2 and phot2 were found to be specifically required for the stability of the R protein HRT and promoting resistance to Turnip Crinkle Virus (TCV) in Arabidopsis [82]. On the other hand, CRY-interacting bHLH (CIB1) has been shown to inhibit immunity in response to different PAMPs [83,84]. This highlights that plants use BL to finely balance resources distributed to promote photosynthesis, growth, or immunity to achieve optimal fitness imposed by environmental limitations.

#### 2.4.3. Zeitlupe Family

The F-box proteins Zeitlupe (ZTL), flavin-binding, Kelch repeat, F-box1 (FKF1), and LOV Kelch Protein2 (LKP2) are a group of BL photoreceptors that contain a LOV domain, an F-box motif, and a Kelch repeat domain. The LOV domain, like those of phots, bind flavin mononucleotide; the F-box motif is involved in the formation of the E3 ubiquitin ligase SCF complex; and the Kelch repeats form a β-propeller structure, associated with protein–protein interactions [85]. ZTL is a component of the circadian clock, regulates hypocotyl elongation in high temperatures, and also regulates ABA-induced stomatal closure [86] (Figure 1). Plants control stomatal aperture to balance water availability with gas exchange and photosynthesis. This is partly regulated by the circadian clock, but also by light and stress responses. Many plant pathogens use stomata as a point of entry and exit and often manipulate the host to keep stomata open; however, this can facilitate the release of volatile stress signaling to warn neighboring plants of attack [87]. It appears that ZTL is a link between the circadian clock and regulation of JA-mediated defenses. One study has shown that RNA interference of *ZTL* in wild tobacco (*N. attenuata*) plants dramatically affects the root circadian clock, reducing the expression of nicotine biosynthetic genes and attenuating resistance to the African cotton leafworm *Spodoptera littoralis* which is a pest of many cultivated crops [88]. These authors’ findings suggest that ZTL regulates JA signaling by direct binding of JAZ proteins and consequently regulating the JAZ-MYC2 module required for nicotine biosynthesis. It was reported recently that light-activated FKF1 represses the multifunctional E3 ligase activity of COP1 by inhibiting homodimerization [89]. This means that this family of BL receptors joins the list of light receptors that also disrupt the COP1 complex to induce signaling [90].

### 2.5. UV Radiation and Immunity

Although most plants aim to maximize exposure to sunlight for photosynthesis, this makes them vulnerable to UV radiation that can potentially damage DNA, proteins, lipids, and membranes. However, the mechanisms of perception, signaling, and metabolic pathways triggered or stimulated by UV light treatments are not fully understood [91]. UVR8 has a role in UV-B and UV-C perception and signaling for the induction of defenses against photodamage. In high UV-B, lipid damage or peroxidation by ROS generates linolenic acid oxidation products which serve as JA precursors. Consequently, UV-B radiation enhances defense responses through JA-dependent and JA-independent signaling mechanisms [4,42]. In solanaceous species, UV-B-induced JA signaling can also induce the expression of defense-related proteinase inhibitors (PIs) that prevent herbivory [92]. In multiple ways, UV-B perception can deter and increase tissue toxicity in some insect herbivores [4,42,93]. In addition to controlling insect pest populations, there is evidence that activation of UVR8 promotes synthesis of phenolic compounds that defend against plant pathogens. Moreover, the energy provided by UV light stimulates the production of potential immunity-related chemicals such as ROS produced by chloroplast photosynthetic machinery, membrane-localized NADP(H) oxidase activity, and the superoxide-generating peroxisomal xanthine oxidase activity [94]. On the other hand, two Arabidopsis protease inhibitors, serpins (AtSRP4 and AtSRP5) involved in biotic stress responses, are upregulated upon UV treatments and challenge with avirulent pathogens as negative regulators of cell death to limit tissue damage [95]. Furthermore, UV-B radiation is directly harmful to some plant pathogens, e.g., reducing spore viability from some plant pathogenic species of fungi and oomycetes [94,96]. 

## 3. Light Signaling and Organelles Targeted by Pathogens

### 3.1. Effectors Targeting Light Signaling Pathways

As shown in the previous section, there is complex interplay between light, hormone, and defense signaling. Unsurprisingly, pathogen effectors (Table 1; Figure 2) have been demonstrated to target canonical light signaling pathways.

The RXLR effector Pi06099 from *P. infestans* is nucleo-cytoplasmically localized and promotes pathogen colonization [101]. Pi06099 interacts with the RL receptor phyB from both Arabidopsis and potato in Y2H and by CoIP (Figure 2) [99]. Nothing further is known about how this interaction affects RL signaling, although *phyB* mutant plants are more susceptible to a variety of pathogens and display reduced SA and JA signaling [113]. Moreover, plants grown under white light and then exposed to additional FRL show increased susceptibility to several pathogens [114]. However, *phyB* mutants are more resistant to sheath blight in rice [66]; therefore, the situation is likely to be complex.

HaRxL106 from oomycete *Hyaloperonospora arabidopsidis* interacts in the nucleus with transcriptional co-regulator radical-induced cell death1 (AtRCD1) (Figure 2) [104]. Plants expressing this effector have both photomorphogenic and reduced SA signaling phenotypes. AtRCD1 interacts with, and is likely phosphorylated by, a group of Mut9-like kinases (MLKs) that were also shown to alter SA-mediated gene expression [104]. The MLKs are photoregulatory protein kinases (PPKs), phosphorylate PIF3, and photoreceptor CRY2 [115,116]. HaRxL106 requires intact RCD1 to mediate its phenotypes. It is reported to use RCD1 activity as a transcriptional co-regulator interfering with light and defense signaling [104]. In contrast, effector HaRxLL470 interacts with the well-known light signaling component HY5, a bZIP TF (Figure 2). HaRxLL470 reduces the DNA binding of HY5 in order to suppress the induction of defense genes [106]. This mechanism may be conserved as a homologous effector from *P. infestans* Pi09585 and was also shown to interact with HY5 from *N. benthamiana* [106].

The effector AVR2 from *P. infestans* manipulates crosstalk between hormone/light and immunity signaling, promoting growth at the expense of defense. AVR2 interacts with BSU-like (BSL) phosphatases resulting in enhanced BR signaling upregulating transcription factor StCIB1/HBI1-like1 (StCHL1) [84,102]. StCHL1 is activated by BR and BL through interaction with CRYs and was found to negatively regulate immunity (Figure 2), promoting pathogen colonization. It is correspondingly downregulated upon PAMP perception [84,102]. *P. infestans* effector Pi02860 interacts with StNRL1, a downstream transducer of BL signaling [97]. Pi02860 enhances the ability of StNRL1 to target the positive regulator of defense, SWAP70, for degradation by the 26S proteasome (Figure 2) [73]. More recently, it was demonstrated that the perception of BL by StPhot1 suppresses plant defenses and also enhances StNRL1’s ability to target SWAP70 for degradation [75]. Moreover, Pi02860 also interacts with 14-3-3 phosphobinding proteins in potato and Arabidopsis (Figure 2) [99]. The 14-3-3 proteins may have many pleiotropic functions but also interact with NRL proteins to modulate their activity following BL perception [98]. However, 14-3-3 proteins have been recently demonstrated to promote the degradation of PIF3 through interactions with activated phyB and MLKs [100], showing that RL and BL share certain signaling components.

It appears that the effectors which target light signaling enhance or utilize natural regulatory crosstalk points in the plant where light signaling suppresses defense signaling to tip the balance in the pathogen’s favor (Figure 2).

### 3.2. Effectors Localizing to the Chloroplast

More than 30 effector proteins from bacteria, fungi, and oomycetes have been identified to localize within or at the chloroplast. For some of these effectors, the host targets remain unknown, and these will not be covered in detail (Table 2). However, for other chloroplast-localized effectors there are some commonalities in the target proteins (Table 2, Figure 3).

#### 3.2.1. Photosystem II

Photosystem II (PSII) is highly targeted by effectors from all kingdoms (Figure 3). HopR1 and HopBB1 from *Pseudomonas syringae* were part of a large Y2H screen conducted by Mukhtar and colleagues with no follow-up studies. However, the identified target of both HopR1 and HopBB1 is PTF1, which is a TF that regulates PsbD (Figure 3) [121]. PsbD encodes protein D2, while PsbA encodes D1 of PSII’s reaction center. PsbD and PsbA bind the redox-active cofactors involved in energy conversion; therefore, a loss of PsbD would block electron transport, resulting in destabilization of the PSII complex [120]. In addition, the effector Sntf2 from the fungal pathogen *Colletotrichum gloeosporioides* interacts with the PSII assembly factor, Mdycf39 (Figure 3) [138]. Mdycf39 has high homology to Ycf39 from the cyanobacterium *Synechocystis*, which is involved in the synthesis of the D1 subunit (PsbA), suggesting that Mdycf39 is important for the structure and function of PSII [138,142,143]. Apple *Mdycf39*-silenced transgenics showed serious growth phenotypes with a loss of chlorophyll. However, *Mdycf39* overexpression lines showed no phenotypic changes in growth, but they had increased susceptibility when infected with WT *C. gloeosporioides* [138]. Additionally, WT isolates showed less H_2_O_2_ and callose production compared to the deletion mutant *∆sntf2-1*. Therefore, Sntf2 can impede the generation of H_2_O_2_ and callose [138].

The reaction center of PSII is targeted by fungal and bacterial effectors, indicating a convergent approach to mediating immune signals. *P. syringae* effectors HopR1 and HopBB1 interact with the TF PTF1, which regulates PsbD, while Sntf2 targets Mdycf39 which is involved in the synthesis of PsbA (Figure 3) [121,138]. One would speculate that the intended target of these three effectors is the regulation of the PSII reaction center, which is involved in energy conversion and electron transport, thereby affecting the amount of ROS induced by the chloroplast.

HopN1 from *P. syringae* has cysteine protease activity. Therefore, when it binds to its target protein, PsbQ from Photosystem II, it degrades PsbQ, resulting in a reduction in electron transport, cROS, and O_2_ production (Table 2, Figure 3) [123]. PsbQ is one of three oxygen-evolving enhancer proteins (OEEs) (OEE1 (PsbO), OEE2 (PsbP), and OEE3 (PsbQ)) which bind to the periphery of PSII [144]. PsbQ and PsbP coordinate the function of the donor and acceptor sides of PSII and stabilize the active form of the PSII-light-harvesting complex II (LHCII) supercomplex [145]. The loss of PsbQ results in reduced assembly and stability of PSII in low-light growth conditions [146]. RXLR31154 from *Plasmopara viticola* localizes to the cytoplasm, nucleus, and chloroplasts, despite not containing a predicted chloroplast transit peptide [141]. Within the chloroplast, it interacts with *Vitis piasezkii* PsbP (Figure 3) which also regulates the water-splitting reactions and is critical for the assembly and stability of the functional core of PSII [147]. The loss of PsbP impairs the accumulation of active forms of PSII supercomplexes and therefore photosynthesis [147]. *P. viticola* infection on a WT plant resulted in VpPsbP accumulation as RXLR31154 stabilizes the PsbP protein. In accordance with this, plants overexpressing PsbP showed increased infection, while plants with reduced expression of PsbP had stunted growth and bleached leaves but reduced infection [141], indicating that PsbP acts as a susceptibility factor, as has been described for the targets of several RXLR effectors [148]. RXLR31154 does not affect the chlorophyll fluorescence capability of plants. However, it does suppress H_2_O_2_ production and INF1-triggered cell death while activating ^1^O_2_-mediated signaling [141]. The ability of RXLR31154 to inhibit INF1-induced cell death is reduced in plants that have been silenced for *PsbP*, suggesting that PsbP is required for this inhibition [141]. Convergent targeting of two proteins of the extrinsic subunits of PSII hints towards a similar virulence strategy across the pathogenic kingdoms.

Another effector that targets PSII is ToxA. This effector was first identified in *P. tritici-repentis* but has also been identified in two other wheat necrotrophic pathogens, *Parastagonospora nodorum* and *Bipolaris sorokiniana* [149]. The wheat chloroplast target of ToxA is ToxA-Binding Protein 1 (ToxABP1) [128] (Figure 3), which has sequence homology with Thylakoid formation 1 (Thf1) from Arabidopsis. Thf1 regulates the PSII-light-harvesting complex (LHCII) interaction and dissociation. The dynamics of PSII-LHCII interaction are essential for the correct control of repair mechanisms; for example, the disassembly of PSII-LHCII is essential for chlorophyll degradation and PsbA repair [150]. If not correctly monitored, this could lead to non-functional PSII complexes and an increase in ROS production. ToxA induces necrosis in sensitive wheat lines that carry the *Tsn1* gene. This gene encodes a classical NB-LRR protein [151]. ToxA-induced necrosis in *Tsn1* wheat lines requires light to occur and results in the accumulation of ROS [128,152], indicating that these are derived from the chloroplast. ToxA expression in *Tsn1* wheat lines also results in disruption of thylakoids, a decrease in PSII, and a loss of chlorophyll [128,153]. 

Interestingly, Thf1 is not only targeted by ToxA but affects *P. syringae* and viral infection. The silencing of tomato and Arabidopsis *Thf1* orthologues results in faster lesion development during *P. syringae* infection [154]. Additionally, these authors showed that the chloroplast localization or stability of GFP-ALC1 was altered by the application of coronatine, a phytotoxin that mimics the plant hormone jasmonic acid isoleucine [154]. Thf1 has also been shown to interact with the coil–coil domain of three *Solanaceae* CC-NLR proteins, N’ from *N. sylvestris* (recognizes the coat protein from Tomato Mosaic Virus (ToMV)), R3a from potato (recognizes AVR3a from *Phytophthora infestans*), and L^3^ from pepper (recognizes the coat protein of Tobacco Mosaic Virus (TMV) and ToMV) [155]. Upon R protein recognition and activation, the resulting cytoplasmic interaction of Thf1 with N’/R3a/L^3^ reduces the stability and therefore levels of Thf1 [155]. This affects chloroplast homeostasis and the induction of a light-dependent hypersensitive response [155].

#### 3.2.2. Calcium-Sensing Receptor and SA Signaling

Another common target of multiple effector proteins is the calcium-sensing receptor (CaS), which regulates SA signaling and is involved in both PTI and ETI against the bacterial pathogen *P. syringae* [156]. The effector SsITL from the necrotrophic fungal pathogen *Sclerotinia sclerotiorum* interacts with CaS resulting in a reduction in SA accumulation early in an infection (Figure 3) [130]. Arabidopsis lines overexpressing CaS showed no growth phenotype but had enhanced resistance to *S. sclerotiorum*, indicating that CaS-mediated resistance relies on Ca^2+^ signaling [130]. Plants overexpressing SsITL showed no phenotypic changes but had reduced resistance to *S. sclerotiorum* [130]. RipG1, also known as GALA1, from bacterium *Ralstonia solanacearum*, has been shown to localize to the plasma membrane and chloroplast concurrent with it containing both an N-myristoylation motif and a cTP [126]. The chloroplast localization of RipG1 increases upon flg22 treatment of leaves and results in a reduced calcium burst (Figure 3) [126]. Plants expressing a chloroplast-targeted RipG1 (RipG1_G2A_) showed reduced SA-responsive gene expression and increased susceptibility to *P. syringae* [126]. In addition, these authors showed that the C4 protein from Tomato Yellow Leaf Curl virus also contains an N-myristoylation motif and a cTP and relocates from the plasma membrane to the chloroplast during the activation of defense by PAMPs [126]. Upon relocalization, C4 interacts with the chloroplast-localized CaS and suppresses its function, including the reduction in SA-responsive gene expression and SA production [126]. Additionally, the authors used plant proteome data from Arabidopsis, tomato, and rice to determine 26 core plant proteins which all contain an N-myristoylation motif and a cTP site. Arabidopsis Calcium-Dependent Kinase 16 (CPK16) was shown to undergo the same PM to chloroplast relocalization during the activation of defense. However, this plant protein induces the expression of SA-responsive genes and results in reduced infection by *P. syringae* [126]. These authors have shown that an innate plant pathway, in which proteins translocate from the plasma membrane to the chloroplast and use CaS to mediate Ca^2+^ and SA signaling, has been co-opted by both viral and bacterial pathogens to reduce resistance. HopI1 from *P. syringae* interacts in yeast with heat shock protein 70 (Hsp70), but this has not been confirmed in planta [117]. This effector also reduces SA and alters thylakoid structure (Figure 3) [117]. 

#### 3.2.3. Cytochrome b6-f Complex

The wheat yellow rust effector 12806 (Pst_12806) from *Puccinia striiformis* f. sp. *Tritici* (*Pst*) interacts with the wheat protein TaISP (Figure 3) [137]. TaISP is a putative subunit of the cytochrome b6-f complex which is involved in the electron transport between PSII and PSI [137]. The loss of this gene in *N. benthamiana* limits the electron transport rate and CO_2_ assimilation rate, while in wheat it results in increased resistance to *Pst* [137,157,158]. Pst_12806 is required for the full virulence of *Pst* on wheat as the expression of Pst_12806 reduces the electron transport rate, photosynthesis, and production of chloroplast-derived ROS, indicating that it suppresses basal immunity and PTI-induced gene expression [137].

#### 3.2.4. JA Signaling

Three effectors from *R. solanacearum* have been identified to localize to the chloroplast, RipAL, RipAD, and RipG1 (Figure 3) [124,125,126,159]. RipAL contains a putative lipase domain which has sequence similarity to the Arabidopsis protein DEFECTIVE IN ANTHER DEHISCENCE1 (DAD1) which catalyzes the release of linoleic acid, a precursor of JA biosynthesis, from chloroplast membranes [124,160]. In planta expression of RipAL increases JA and JA-isoleucine levels while suppressing SA levels. This hormone regulation was abolished when DAD1-like mutant RipALs (RipAL^S257A^ and RipAL^D329A^) were expressed. However, these mutations did not alter the chloroplast localization of the effectors [124]. RipAL is also able to suppress PTI responses such as ROS burst and marker gene expression, while the mutants could not [124]. RipAD was in silico predicted to contain a nuclear localization signal (NLS). However, transient overexpression in *N. benthamiana* leaves shows it has cytosolic and chloroplastic localization [125]. This effector significantly reduced the PTI-induced ROS burst, presumably from the chloroplast (Figure 3) [125] and potentially by inhibiting cROS production. 

### 3.3. Effectors Which Prevent Their Targets Localizing to Chloroplasts

Above, we examined the role of effectors that localize within or around the chloroplast. This section explores effector proteins that do not localize to the chloroplast but interact with chloroplast-targeted proteins (Table 1; Figure 2). 

Two effectors, Pst_4 and Pst_5, from the fungal stripe pathogen *Puccinia striiformis* f. sp. *Tritici* prevent the wheat chloroplastic TaISP protein from being transported to the chloroplast, resulting in reduced ROS and immunity (Figure 2) [111]. As discussed above, Pst_12806 from the same pathogen also interacts with TaISP but inside the chloroplast to reduce electron transport (Figure 3), suggesting that this protein is an important target for *Puccinia* as it employs multiple strategies to disrupt its activity. In addition, AVRvnt1 from oomycete *P. infestans* decreases the accumulation of glycerate 3-kinase (GLYK) in the chloroplast. AVRvnt1 binds to the full-length GLYK protein but not to the shorter form lacking the chloroplast transit peptide. The recognition and activation of the plant resistance gene Rpi-vnt1 requires both GLYK and light [112].

HopAU1 from bacterium *P. syringae* pv. *Actinidiae* has been shown to interact with the chloroplastic CaS (Figure 2) [107]. HopAU1 causes cell death in *N. benthamiana*, and the silencing of CaS reduces the HopAU1-induced cell death, indicating that CaS could be important in initiating the immune response [107]. Interestingly, CaS is also a conserved target of other pathogen effectors, fungal SsITL, viral C4, and possibly bacterial RipG (Figure 3), indicating the importance of chloroplastic Ca signaling to different types of pathogens. As HopAU1 was found to localize to the cytoplasm [108], perhaps HopAU1’s function is to prevent CaS localizing to chloroplasts (Figure 2). Taken together with Pst_4, Pst_5, and AVRvnt1, this would suggest that preventing chloroplast localization of nuclear-encoded chloroplast gene products is a conserved mechanism employed by different classes of plant pathogens. 

In contrast, the RipG effectors from *R. solanacearum* are thought to function as components of an E3 ubiquitin ligase complex and interact with Skp1 by mimicking F-box proteins [110]. RipG2 and RipG7 were shown to interact with Skp1 and the nuclear-encoded chloroplast proteins Cab13, RbcX, and RbcS (Figure 2). These proteins may be targeted for 26S proteasome degradation to suppress plant defence [109]. Cab13 is a light-harvesting chlorophyll a/b-binding protein and part of PSII, while RbcX and RbcS are chaperones involved in RuBisCo assembly and the light chain of RuBisCo, respectively [109]. The different components of PSII seem to be heavily targeted by pathogen effectors (Figure 3). Altogether, this would suggest that: (1) chloroplast processes are important for plant resistance to pathogens, and (2) preventing chloroplast localization of nuclear-encoded chloroplast proteins can occur by blocking transit or by causing protein degradation.

## 4. The Impact of Light on Pathogens

### 4.1. Light Perception by Pathogens

Of course, light is not only perceived by plants; pathogens, pests, and microbes have their own photoreceptors and can sense and respond to light in different ways. Plants, bacteria, and fungi sense BL through light–oxygen–voltage (LOV) domains containing proteins [2]. In contrast to plant phototropins, which contain two LOV domains fused to a C-terminal Ser-Thr kinase, in plant-associated bacteria a LOV domain is commonly linked to a His kinase domain [2]. In fungi, the white collar complex (WCC) is involved in BL perception. This consists of two DNA-binding proteins, one of which contains a LOV domain. These act together to form a transcription factor [161]. Much less is known about oomycete photoreceptors, but *P. infestans* is only predicted to contain cryptochromes, which are a class of BL receptor found throughout all kingdoms of life including bacteria and fungi [162,163].

Red light in fungi and bacteria is perceived by phytochromes, although fungal phytochromes (Fphs) and bacteriophytochromes (Bphs) are more closely related to each other than they are to plant phytochromes and contain biliverdin IV instead of phytochromobilin [44]. Moreover, phytochromes from fungal and bacterial plant pathogens are more sensitive to FRL than those from plants, probably due to the relative abundance of FRL inside plant tissues [2]. Light signaling is known to regulate many developmental processes in microbes such as multicellularity, motility, reproduction, and germination; however, more recent work is shedding light on the roles of light in pathogenicity [2,163].

### 4.2. Impact of Light on Pathogen Growth, Sporulation, and Virulence

In *P. infestans*, light was found to delay sporulation so that it occurs during the night, which is thought to protect spores from harmful exposure to UV light. This signaling was found to involve Myb TFs whose expression levels change with light/dark cycles [162]. In *H. arabidopsidis*, both constant light and constant dark suppressed sporulation, whereas spore germination, mycelial growth, and oospore formation were stimulated by constant light but not constant dark [164]. Infections carried out at dusk were also more successful than those started at dawn, but it is uncertain how much this has to do with the light regulation of virulence mechanisms in pathogens versus light regulation of defense mechanisms in the host [164]. 

Red light seems to negatively regulate virulence in some bacteria. *Xanthomonas campestris* pv. *Campestris* (*Xcc*) possesses a bacteriophytochrome (XccBphP). Upon light exposure, XccBphP downregulates the production of xanthan exopolysaccharides and biofilm formation, and this corresponds to a reduction in bacterial virulence. Plants infected with XccbphP mutants accumulate less callose and show more stomatal opening than those in response to WT infection [165]. Additionally, studies have shown that BL seems most important in the very early stages of infection for *P. syringae* and leads to the upregulation of chemotaxis receptors, two of which were shown to be required for full virulence. Interestingly, both blue (PsPtoLOV) and red (PstBphP1) photoreceptors were required for transcriptional changes to BL, suggesting that PstBphP1 also responds to BL [166]. However, a similar study found distinct roles for different types of light. Blue light caused the upregulation of T3SS genes involved in virulence, while RL suppressed the production of coronatine, a bacterial compound known to open plant stomata. Indeed, Pstbph mutants showed increased stomatal opening in plants compared to WT infection [167]. This suggests a complex interplay between different light receptors and that crosstalk between signaling pathways happens in pathogens as well as in plants. 

In the fungus *Alternaria alternata*, BL was found to repress spore generation but stimulate the production of the mycotoxins alternariol (AOH) and altertoxin (ATX). Mutants in the BL receptor WC-1 gene lreA no longer repressed spore formation, while ATX was also induced in the dark [168]. Interestingly, RL can reverse the BL phenotypes, suggesting interplay between blue (LreA) and red (FphA) photoreceptors, with both involved in activating stress-related MAP kinase signaling through HogA [169]. *Botrytis cinerea* is a broad-host-range fungal pathogen, and its RL receptor BcPhy3 has been demonstrated to be important for normal growth development with mutants exhibiting impaired growth, reduced cell wall integrity, and decreased virulence on several plant hosts [170]. This suggests that RL perception is important to *Botrytis*. Intriguingly, recent work has shown that RL and FRL perception in plants regulates defense against *Botrytis*, specifically by reducing the expression of *Botrytis*-responsive defense genes and delaying recognition and JA signaling [171]. However, another study has shown that red LED light improved basal resistance to *Botrytis* isolates [172]; therefore, more research is needed to unravel the complexity involved.

### 4.3. Light as Pest Control

Light can be used as a weapon to reduce plant infection or even kill phytopathogens such as *Botrytis* as an alternative to using fungicides which have potential harmful effects on human health. Treatment with UV-C light followed by 4 hours in the dark was able to kill *Botrytis* while the plants were not negatively impacted by the same treatment [173]. Similar work demonstrated that a night-time treatment of UV-B light significantly reduced the severity of powdery mildew infections in strawberry and rosemary [96]. In general, UV light applications have a positive effect on disease resistance, mainly through the increase in metabolites and through directly damaging effects on species with life stages vulnerable to UV radiation [92]. There are already exciting advances in the use of robot supplemental UV delivery systems in horticultural crops to control fungal and oomycete pathogens pre-harvest [91,174]. It will be exciting to see if UV treatments can be applied more widely to other field crops to reduce reliance on chemical fungicides. The ability to manipulate or control pathogens and propagate plants primed for appropriate plant immunity and environmental resilience through light treatments is particularly relevant to greenhouses and vertical farming systems where there is huge potential to manipulate light spectra easily.

## 5. Conclusions and Future Perspectives

It is clear that there are many points of crosstalk between light signaling and plant immunity and that pathogens have evolved to manipulate key components in order to benefit themselves. Pathogen effectors target light signaling pathways to exploit crosstalk points where light negatively regulates defense. Effectors targeting chloroplasts often converge on the central processes, PSII and Ca^2+^ signaling. Indeed, non-chloroplast-localized effectors may block these processes by preventing the localization of associated chloroplast proteins encoded by nuclear genes. There will be interesting future developments in chloroplast stress research and the multiple roles the chloroplast plays in defense, with the potential of discovering new signaling molecules and intersectional components. We will learn a lot about chloroplast immunity from studying how pathogen effectors target chloroplast processes. 

As automated environmental control and light technology continues to improve and become more accessible, there is a huge amount still to learn from decoupling the plant responses to different wavelengths of light and about how this could be harnessed to improve plant health in all plants of economic value, whether they are grown indoors or in the field. A greater understanding of light signaling pathways in plants, their intersection points with immune regulation, and how pathogen effectors manipulate them could provide new avenues to engineer resistance to a wide variety of pathogens given the prevalence of commonly targeted processes. Similarly, as more is learned about light signaling perception in pathogens, can we then manipulate light wavelengths and control plant growth regimes to prevent disease?

## Figures and Tables

**Figure 1 ijms-24-03803-f001:**
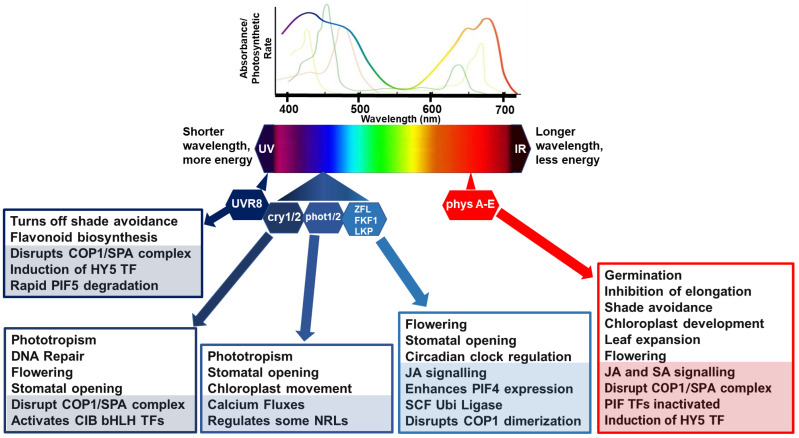
**Simplified model of light perception and associated molecular and physiological processes.** Lines, above the light spectrum, represent the light absorption spectra for the major photosynthetic pigments, chlorophyll *a* (lime green), *b* (apple green) and carotenoids (brown/green line). The rainbow line highlights photosynthetic rate relative to wavelength. Below the light spectra indicates multiple photoreceptors that have evolved to detect blue and red wavelengths. In general, in poor light quality, stabilization of photomorphogenesis-promoting factors occurs, promoting growth in the direction of light. Photomorphogenic responses are regulated by phytochromes (phyA to E), which absorb RL/FRL; cryptochromes (cry 1/2), which are activated by BL/UV-A; phototropins (phot 1/2), activated by BL/UV-A; and Zeitlupe family members (Ztl, fkf1, lkp2) activated by BL/UV-A and the UV-B photoreceptor (UV resistance locus; UVR8). Light activation of phytochromes, cryptochromes, and UVR8 promotes interactions with transcription factors to relay responses to light conditions. CONSTITUTIVE PHOTOMORPHOGENIC 1 and SUPPRESSOR OF PHYTOCHROME A (COP1/SPA) complex act as substrate adaptors for CULLIN4 (CUL4) E3 ligase complexes, key repressors of light responses in darkness. Under high RL:FRL conditions, ELONGATED HYPOCOTYL 5 (HY5) is induced and Phytochrome-Interacting Factor (PIF) activities are inactivated by phyB to suppress shade avoidance responses such as elongation. Under UV and blue lights, COP1/SPA is disrupted to suppress unnecessary and costly photomorphogenesis responses. UVR8 induces HY5 activity, and the binding of UVR8 to COP1 in UV-B disrupts PIF5 stabilization, rapidly lowering PIF5 abundance in sunlight by degradation via ubiquitination and proteasome-mediated protein degradation. CRYs disrupt the COP1/SPA complex which activates cryptochrome-interacting basic helix–loop–helix (CIB1) and CIB1-related proteins to promote floral initiation, DNA repair, and stomatal opening. Phots have their own kinase activities associated with regulation of physiological responses to optimize photosynthesis, such as stomatal opening to regulate CO_2_ uptake and water loss, chloroplast movements, calcium fluxes, and leaf positioning. Activated phots interact with some NRL family members as a substrate adaptor c of a CULLIN3 E3 ubiquitin ligase. Zeitlupe family members disrupt COP1 dimerization, enhance PIF4 expression, and regulate responses such as flowering time, JA biosynthesis, and stomatal opening. COP1/SPA can also target some additional E3 ligases or kinases for degradation causing indirect increases and stabilization of downstream transcription factors, such as PIFs. In each box, the lower shaded sections highlight molecular mechanisms, and the non-shaded, upper sections features physiological processes.

**Figure 2 ijms-24-03803-f002:**
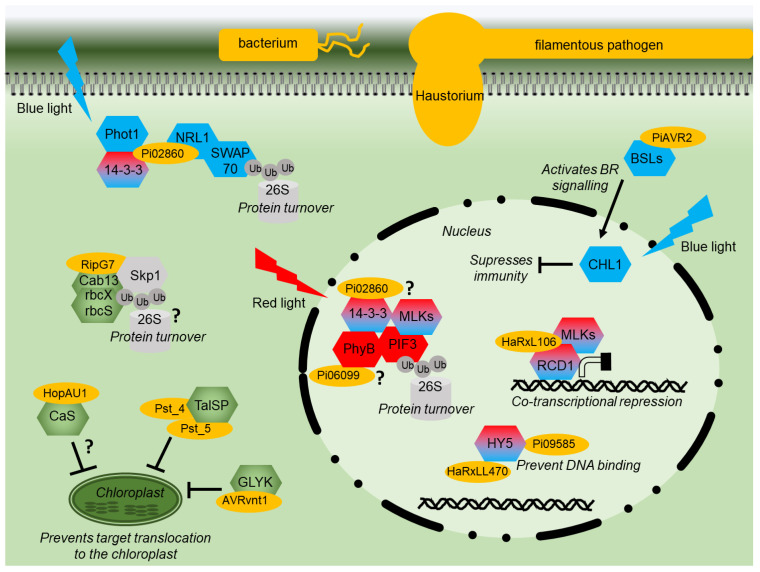
**Effector modes of action diagram.** Diagram of a plant cell showing the modes of action of various pathogen effector proteins (yellow). Signaling components of red or blue light signaling are shown in red or blue, respectively. Shared components are shaded red and blue. Proteins with a chloroplast function are shown in green. Components of the proteasome are indicated in grey. Interactions that remain to be fully characterized are indicated by “?”.

**Figure 3 ijms-24-03803-f003:**
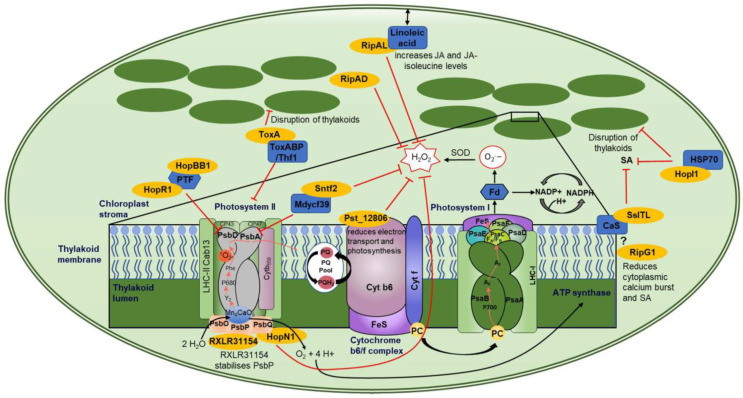
**Chloroplast**-**localized effector proteins and their targets.** This figure shows a representative image of a chloroplast and the effector–target interactions with downstream mode of action. The enlarged box shows a schematic of the thylakoid membrane (dark green) with PSII, cytochrome b6/f complex, and PSI (various colors to better visualize subunits) along with effectors targeting various components (dark blue) of these complexes. Effectors are shown in yellow, inhibitory signaling is shown by a red line, while electron transport is shown by black arrows. The **?** represents speculative interactions, and double-ended arrows represent movement.

**Table 1 ijms-24-03803-t001:** Effectors targeting light signaling pathways and inhibition of protein translocation into the chloroplasts.

Effector	Organism	Effector Localization	Host Target	Mode of Action	References
Pi02860	*P. infestans*	Cytoplasmic	StNRL1	02860 interacts with StNRL1 to activate a BL pathway that suppresses immunity.	[73,75,97]
Pi06099	*P. infestans*	Nucleo-cytoplasmic	St14-3-3, At14-3-3	Pi02860 interacts with 14-3-3s which modulate RL and BL signaling.	[98,99,100]
Pi06099	*P. infestans*	Nucleo-cytoplasmic	StPhyB, AtPhyB	Pi06099 interacts with the RL receptor, phyB. PhyB mutants have reduced SA gene expression.	[99,101]
PiAVR2	*P. infestans*	Nucleo-cytoplasmic	StBSLs	PiAVR2 interaction with BSL phosphatases increases BR signaling activating StCHL1 which suppresses immunity.	[84,102]
HaRxL106	*H. arabidopsidis*	Nucleus	AtRCD1	HaRxL106 interacts with transcriptional co-regulator RCD1 that interacts with MLKs which integrate signaling of blue and red photoreceptors.	[103,104]
HaRxLL470, Pi09585	*H. arabidopsidis*	Nucleo-cytoplasmic	AtHY5, HYH, NbHY5	HaRxLL470 interacts with bZIP TF HY5 and suppresses defense gene induction by disrupting binding of HY5 to DNA.	[105,106]
HopAU1	*P. syringae* pv. *actinidiae*	Cytoplasmic	CaS	HopAU1 interacts with CaS, inhibiting translocation into chloroplast.	[107,108]
RipG2, RipG7	*R. solanacearum*	Cytoplasmic	Nbcab13, NbrbcX, NbrbcS	RipG2 and RipG7 interact with targets for proteosome degradation.	[109,110]
Pst_4, Pst_5	*Puccinia striiformis f.* sp. *tritici*	Cytoplasmic	TaISP	Pst_4 and Pst_5 interact with TaISP, inhibiting translocation into chloroplast.	[111]
AVRvnt1	*Phytophthora infestans*	Cytoplasmic	GLYK	AVRvnt1 interacts with GLYK, inhibiting translocation into chloroplast.	[112]

**Table 2 ijms-24-03803-t002:** Effectors that localize to the chloroplast.

Effector	Organism	Host Target	Mode of Action	References
Bacteria				
HopI1	*P. syringae* pv. *maculicola*	HSP70	HopI1 affects the activity and/or specificity of Hsp70 and induces altered thylakoid structure and reduced SA accumulation.	[117]
AvrRps4	*P. syringae* pv. *pisi*		Suppress ROS production and callose deposition.	[118,119]
HopK1	*P. syringae* pv. *tomato*		Suppress ROS production and callose deposition.	[118,119]
HopO1-2	*P. syringae* pv. *tomato*			[120]
HopR1	*P. syringae* pv. *tomato*	PTF1, CBSX2	PTF1 is a TF that regulates PsbD. PsbD is a PSII reaction center protein, and the loss of PsbD blocks electron transport and destabilizes the PSII complex.	[120,121,122]
HopBB1	*P. syringae* pv. *tomato*	PTF1	PTF1 is a TF that regulates PsbD. PsbD is a PSII reaction center protein, and the loss of PsbD blocks electron transport and destabilizes the PSII complex.	[120,121,122]
HopN1	*P. syringae* pv. *tomato*	PsbQ	HopN1 interacts with and degrades PsbQ, reducing oxygen production and electron transport and attenuating cROS.	[123]
HopM1	*P. syringae* pv. *actinidiae*			[108]
RipAL	*R. solanacearum*		RipAL induces JA production and suppress SA signaling in plant cells.	[124]
RipAD	*R. solanacearum*		RipAD suppressed flg22-triggered ROS presumably from the chloroplast.	[125]
RipG1	*R. solanacearum*		RipG1 reduces the cytoplasmic calcium burst in response to flg22 treatment.	[126]
Las5315	*Candidatus Liberibacter asiaticus*		Las5315 induces starch accumulation by increasing starch production and reducing starch degradation enzymes.	[127]
**Fungi**				
ToxA	*Pyrenophora tritici-repentis*	ToxABP1 (Thf1)	ToxA causes the disruption of thylakoids, decrease in PSII, and a loss of chlorophyll.	[128,129]
SsITL	*Sclerotinia scleritorium*	CAS (calcium sensing)	SsITL inhibits SA accumulation during the early stage of infection by interacting with CAS.	[130]
RsCRP1	*Rhizoctonia solani*			[131]
CTP1, CTP2, CTP3, MLP124111, Mlp72983	*Melampsora larici-populina*			[132,133,134]
PST03196, PST18220, PstCTE1	*Puccinia striiformis f.* sp. *tritici*			[135,136]
PST12806	*Puccinia striiformis f.* sp. *tritici*	TaISP	PST12806 interacts with TaISP resulting in reduced electron transport, photosynthesis, and production of cROS.	[137]
PGTG_00164, PGTG_06076	*Puccinia graminis f.* sp. *tritici*			[129]
Sntf2	*Colletotrichum gloeosporioides*	Mdycf39	Sntf2 suppresses plant defense responses by reducing callose deposition and H_2_O_2_ accumulation.	[138]
**Oomycetes**				
PhRXLR-C20, PhRXLR-C27	*Plasmoprara halstedi*			[139]
PvRXLR54, PvRXLR161, PvRXLR86, PvRXLR61	*Plasmopara viticola*			[140]
RXLR31154	*Plasmopara viticola*	VpPsbP	RXLR31154 reduces H_2_O_2_ accumulation and activates the singlet molecular oxygen (^1^O_2_) signaling pathway by stabilizing PsbP.	[141]

## Data Availability

Data sharing not applicable. No new data were created or analyzed in this study.

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
