# Peer review of "Tuning the Wavelength: Manipulation of Light Signaling to Control Plant Defense"

_ijms, 2023, doi:10.3390/ijms24043803_

Round 1

Reviewer 1 Report

A brief summary

The review is devoted to the influence of the spectral parts of visible light on the interdependent processes of regulation of growth and immune response in plants and virulence in pests of various species. Effectors targeting light signaling pathways are discussed in detail, especially in chloroplasts. Light perception by pathogens and impact of light on pathogen growth, sporulation and virulence is a separate topic. A better understanding of the effects of light on plant growth and immune activity and on the growth and virulence of pathogens may allow the selection of light conditions that will stimulate plants and inhibit pathogens.

Broad comments 

Nice review on the complex topic of how light affects the plant vs. pathogen system, with both sides being light dependent.

The review contains a large number of little-known abbreviations. It is customary to use an abbreviation if the term occurs three or more times. Therefore, for example, the abbreviation WCC occurring once in line 634 is meaningless. And so on. It is highly recommended to include a list of used abbreviations.

Specific comments 

Line 21: plant, light, immunity, pathogen effectors

Line 126: the decoding of BL and RL must be done in line 126 and not in line 137

Author Response

First of all we would like to thank the reviewers for taking the time to read our review and for the suggestions to help improve it’s quality.  We were pleased to see that overall, the reviewers enjoyed our review and that there were only minor suggested changes to address.

We have addressed all of Reviewer 1’s suggestions as follows:-

Reviewer 1 suggests “It is highly recommended to include a list of used abbreviations” We weren’t sure how to implement this in the draft so we have prepared a separate abbreviation list as a word document.  Specific comments were “

Line 21: plant, light, immunity, pathogen effectors

Line 126: the decoding of BL and RL must be done in line 126 and not in line 137”. To address these points we have track changed the suggested corrections to the key words and the abbreviation at the first use of Blue light and red light as directed by the Reviewer 1.

Thanks again for your contribution,

yours sincerely, Eleanor Gilroy

Reviewer 2 Report

The Manuscript entitled "Tuning the wavelength: Manipulation of light signaling to con- 2 trol plant defense" is well written and informative for young researcher. 

1.  The quality of figure 1 is not good.Please provide the good quality figure.

2. Keyword are  not looking impressive. Please revise the keywords

3. Please check line 35, please check the references

4. Please check the line 124, the format should be according to journal style standred.

5. I have found some typing errors, please check the whole manuscript. For example, please check line 298

6. What do you means of Title 2 and title 3, Please explain clearly  Title 1

and title 2 in tables.

7. What have authors found the innovative ideas from this study?

Author Response

First of all we would like to thank the reviewers for taking the time to read our review and for the suggestions to help improve it’s quality.  We were pleased to see that overall, the reviewers enjoyed our review and that there were only minor suggested changes to address.

We have addressed Reviewer 2’s suggestions where it was clear to us of the mistake:-

1)   Reviewer 2 said “ the quality of figure 1 is not good. Please provide the good quality figure”. We have addressed this by enlarging the text and will upload a better quality image.

2)  We were not sure what Reviewer 2 meant by making key words look more “impressive” as we were using commonly used search terms in our field but we have added some additional key words, " signaling; pathogenicity"

3) “Please check line 35” we have done this and edited.

  1. Please check the line 124, the format should be according to journal style standred” We have also done this.
  2. I have found some typing errors we have checked through the document again. Many typos were due to UK vs US English but have also corrected some sentences which may have been over or under edited and lost good grammer or clear meaning.
  3. What do you means of “Title 2 and title 3”, These are remnants from the manuscript template. We are sorry about this formatting error that has happened while copying table information from excel into the journal’s template file. We struggled repeatedly to get the format right, but I think we have managed now. We are happy to send the original excel files for the journal’s editorial team to upload if there are continuing problems.
  4. What have authors found the innovative ideas from this study?” we believe we have addressed this in our conclusions section as far as we can for a review of published work.

Thanks again for your contribution,

yours sincerely, Eleanor Gilroy